# Topologically trivial gap-filling in superconducting Fe(Se,Te) by one-dimensional defects

A. Mesaros [1], G. D. Gu[2] & F. Massee [1] ✉

Structural distortions and imperfections are a crucial aspect of materials science, on the macroscopic scale providing strength, but also enhancing corrosion and reducing electrical and thermal conductivity. At the nanometre scale, multi-atom imperfections, such as atomic chains and crystalline domain walls have conversely been proposed as a route to topological superconductivity, whose most prominent characteristic is the emergence of Majorana Fermions that can be used for error-free quantum computing. Here, we shed more light on the nature of purported domain walls in Fe(Se,Te) that may host 1D dispersing Majorana modes. We show that the displacement shift of the atomic lattice at these line-defects results from sub-surface impurities that warp the topmost layer(s). Using the electric field between the tip and sample, we manage to reposition the sub-surface impurities, directly visualizing the displacement shift and the underlying defect-free lattice. These results, combined with observations of a completely different type of 1D defect where superconductivity remains fully gapped, highlight the topologically trivial nature of 1D defects in Fe(Se,Te).

Structural defects have seen a resurgence of interest following the realisation that their own topological nature allows them to host topologically protected in-gap states[1], i.e., states that are robust to perturbations other than those that break the symmetry of the system. This insensitivity includes most types of random disorder, and, as such, makes topologically protected states a promising platform for quantum information technology[2,3]. The most obvious structural defect where topologically protected states may be found is the boundary between the material and the vacuum: the end-points of a 1D chain[4–10], the edge of a 2D island[11–15], or the surface of a 3D bulk[16]. Topologically protected states have been predicted and evidenced also on point-like, line-like, and surface-like structural defects within the material or at their intersections with material boundaries, such as dislocations[17], disclinations[18], grain boundaries[19], stacking faults[20], and step edges[15]. More recently, higher-order topology was discovered as a way to protect states at lower dimensional features of structural defects, such as corners on a surface[21–23], making a detailed understanding of the boundary essential.

In the same way as a boundary with vacuum, the boundary between two materials may provide suitable conditions for Majorana modes to exist[9,24]. For a junction between two superconductors with a $\pi$-phase difference, it was shown that dispersive Majorana modes can exist[24]. Recently, a relatively constant in-gap density of states on a structural defect line in crystalline Fe(Se,Te) was reported as evidence of such modes[25]. This raises three important questions: (1) does a relative displacement occurring across a structural defect line induce a phase shift in the superconductor, (2) are the exact direction and magnitude of the structural displacement crucial for the appearance of the in-gap states, i.e., do small perturbations to the displacement lift the topological protection, and 3) what happens at the endpoint of a defect line.

In this work, we measure single crystalline $FeSe_{0.45}Te_{0.55}$ with our 300 mK scanning tunnelling microscope[26] and apply a tailored lattice-phase-shift analysis in order to address these questions and to uncover the exact origin of the structural displacement shift. We find that the

[1]Université Paris-Saclay, CNRS, Laboratoire de Physique des Solides, 91405 Orsay, France. [2]Condensed Matter Physics and Materials Science Department, Brookhaven National Laboratory, Upton, NY 11973, USA. ✉e-mail: freek.massee@universite-paris-saclay.fr

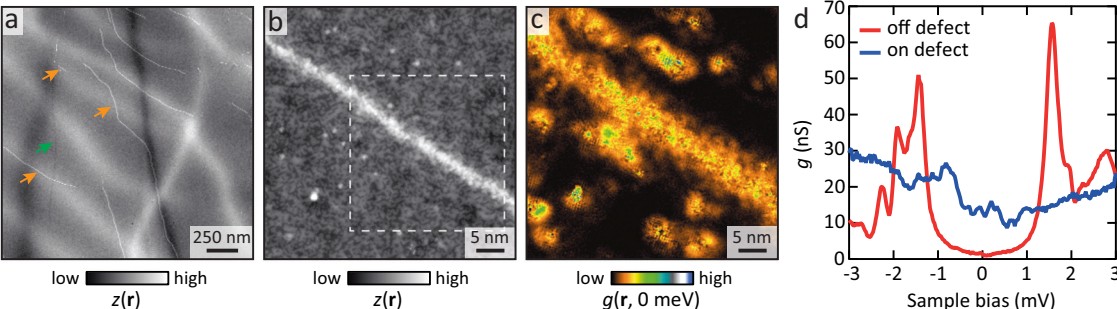

**Fig. 1 | 1D defects in FeSe$_{0.45}$Te$_{0.55}$. a** Large field of view constant-current image of FeSe$_{0.45}$Te$_{0.55}$ containing a number of 1D defects (three are indicated by orange arrows). The green arrow marks a narrow dip that runs roughly vertically. Broader height variations likely reflect strain of the surface. $V = 50$ mV, $I = 50$ pA. The image has been corrected for several minor tip changes. **b** Constant current image on one of the 1D defects in **a**. $V = 5$ mV, $I = 100$ pA. The region in the dashed box is analysed in Fig. 2. **c** Differential conductance, $g$, at $E = 0$ on the area of **b**. **d** Typical spectra taken on the 1D defect and several tens of nanometres away from it: on the defect, the sub-gap states completely fill the gap.

lattice shifts across the defect are typically near $\pi/2$ instead of quantised at $\pi$, making the quantised $\pi$-shift of the superconductor phase unlikely. Further, we find that the in-gap density of states becomes gapped on some segments of the defect, even though the lattice displacement shift remains constant on those segments, giving strong evidence against topological protection of the in-gap modes. Based on all our observations, plus the fact that we are able to reposition sections of a 1D defect with the STM tip, a natural explanation for the in-gap density of states and on average $\pi/2$ phase shift is the presence of sub-surface debris that warps the top Fe(Se,Te) layer(s) and induces non-topological gap filling.

## Results

### 1D defects: topography and spectroscopy

Large, atomically flat surfaces were obtained with clear atomic resolution. Additionally, a number of bright 1D defects were observed, ranging in length from several tens to hundreds of nanometres. Interestingly, they do not have a fixed orientation with respect to the atomic lattice, are seen to change direction, and often abruptly terminate, see Fig. 1a. We note that in addition to these sharp 1D defects, broader intensity variations are also observed, which likely reflect a slight strain of the surface[27]. Lastly, a relatively narrow 1D dip defect can be seen (green arrow in Fig. 1a).

Upon closer inspection, the bright 1D defects (orange arrows in Fig. 1a) show identical characteristics to those reported previously[25]: the defects are a continuous protruding line in topography a few nanometres wide, lacking clear atomic contrast, see Fig. 1b. Additionally, the in-gap density of states along the entire length of the defect is enhanced, effectively filling up the gap, see Fig. 1c, d. We note that despite the presence of a small concentration of excess Fe atoms (whose distribution is the same near and away from 1D defects), the tunnelling spectra away from the defect have a well-defined gap with minimal sub-gap filling and gap sizes in agreement with those reported previously[25,28,29].

### Lattice phase shift determination

Next, we discuss the displacement of the lattice upon crossing the 1D defect. With respect to the ideal square lattice, the displacement vector $(u_x, u_y) = \frac{a}{2\pi}(\phi_1, \phi_2)$ of each atom is defined within the periodic unit-cell of sidelength $a$, and is hence represented by two-phase variables $\phi_\alpha \in [0, 2\pi]$, $\alpha = 1, 2$, called the lattice phases. As detailed in refs. [30,31] the smoothed lattice phases $\phi_\alpha(\mathbf{r})$ defined at all positions $\mathbf{r}$ of the topographic image are determined by comparing the atomic contrast in the topograph with a reference lattice. To accurately extract the lattice phase, the choice of reference lattice is critical. This is because if the position $\mathbf{K}_\alpha$ of the Bragg peak of the reference lattice slightly

deviates from the one of the measured lattice, an extracted lattice phase image $\phi_\alpha(\mathbf{r})$ will have a non-zero linear term, i.e., a slope, since $\cos[\sum_\alpha (\mathbf{K}_\alpha + \delta\mathbf{K}_\alpha) \cdot \mathbf{r} + \phi_\alpha(\mathbf{r})] = \cos[\sum_\alpha \mathbf{K}_\alpha \cdot \mathbf{r} + \phi_\alpha(\mathbf{r}) + \delta\phi_\alpha(\mathbf{r})]$, with the linear term $\delta\phi_\alpha(\mathbf{r}) \equiv \delta\mathbf{K}_\alpha \cdot \mathbf{r}$. If the slope term is nonzero for a given choice of reference lattice, one will over- or underestimate the relative displacement of domains (labelled left ($L$) and right ($R$)), that may be present across the 1D defect, see Fig. 2a. An operational definition of the ideal Bragg peak $\mathbf{K}_\alpha$ is therefore that for which there is a vanishing slope of the $\phi_\alpha(\mathbf{r})$ image[32]. Crucially, since the pixel size of the experimental image is generally not commensurate with $a$, the $\mathbf{K}_\alpha$ is not necessarily positioned on the centre of a pixel in the Fourier transform of the topograph[30–32].

### Reference lattice optimisation

To determine the optimal reference lattice, we calculate the lattice phase images $\phi_\alpha^{\mathbf{K}_\alpha}(\mathbf{r})$ for a finely spaced set of values of $\mathbf{K}_\alpha$ covering the experimental $4 \times 4$ pixel area centred on the brightest pixel in the Fourier transform (Fig. 2c). In practice, each phase image is obtained by shifting the Fourier data so that $\mathbf{K}_\alpha$ is at the origin, then applying a low-pass filter and inverse Fourier transforming. As such, the image visualises the local displacements of the measured topography with respect to the reference lattice defined by $\mathbf{K}_\alpha$, exactly as detailed in e.g., refs. [25,31]. The crucial difference with previous work, however, is that we allow for non-integer values of $\mathbf{K}_\alpha$. Then, for each $\phi_\alpha^{\mathbf{K}_\alpha}(\mathbf{r})$ image (Fig. 2d, e), we calculate the total standard deviation $\varepsilon_{\phi\alpha}(\mathbf{K}_\alpha) \equiv (N_L \varepsilon_{\phi\alpha}^L(\mathbf{K}_\alpha) + N_R \varepsilon_{\phi\alpha}^R(\mathbf{K}_\alpha))/(N_L + N_R)$ from the standard deviations $\varepsilon_{\phi\alpha}^{L/R}(\mathbf{K}_\alpha)$ and number of pixels $N_{L/R}$ of the real-space domains $L$ and $R$, respectively (insets of Fig. 2d, e). The minimum value of the total standard deviation $\varepsilon_{\phi\alpha}(\mathbf{K}_\alpha)$ over the set of considered $\mathbf{K}_\alpha$ then determines the optimal $\mathbf{K}_\alpha$ of the reference lattice and its lattice phases (Fig. 2d, e), since a non-vanishing slope term always increases the standard deviation of a function on a domain. The key method change with respect to ref. [32] is that although a considered $\mathbf{K}_\alpha$ is fixed for the entire image, the standard deviations of the two domains $L$, $R$ are separately determined and then added up—without considering the region where the phase jump occurs. This is because we are aiming to determine the periodicity of the ideal lattice, which we assume to be identical on either side of the defect (see also Supplementary Note 2). Including the phase jump $\Delta\phi$ (which we cannot locally extract due to the absence of atomic contrast) would erroneously increase the total slope of the phase across the image by $\Delta\phi/l$, where $l$ is the image length, thereby effectively stretching or compressing the resulting reference lattice with respect to the ideal one. We find that the lattice phases obtained with the ideal reference lattice are uniform

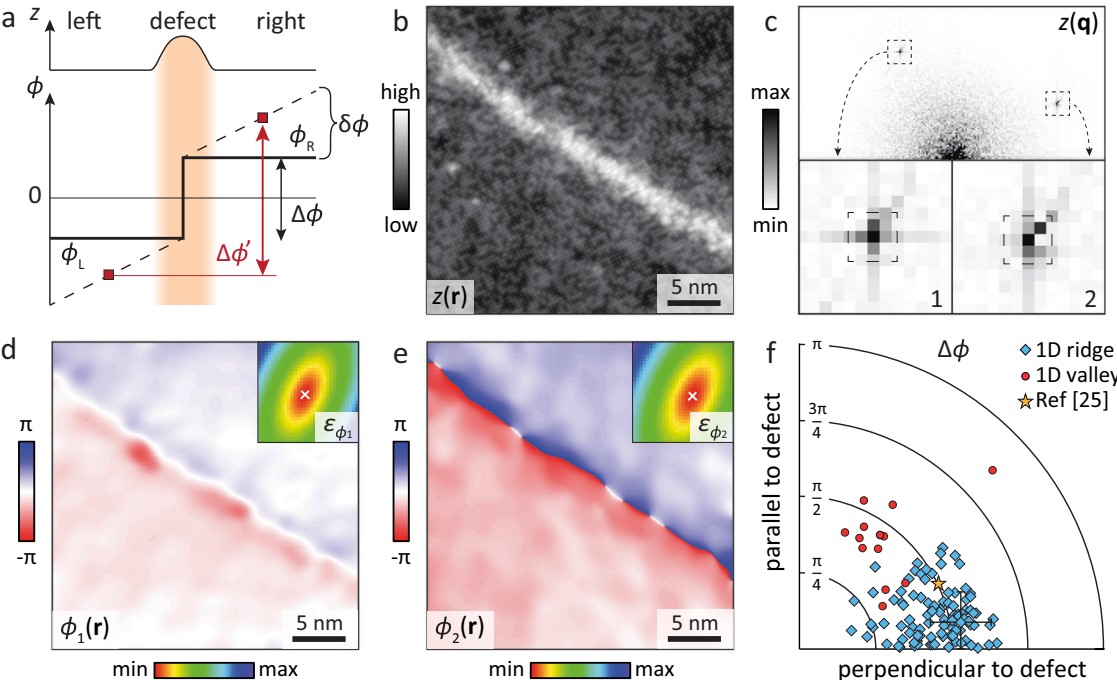

**Fig. 2 | Phase analysis. a** Schematic of the phase extraction method. Since atomic contrast in the defect region itself is blurred (orange shading), it cannot be used to extract the phase jump directly. Any non-zero slope in the phase (dashed line) due to a mismatch between the reference lattice constant and the actual lattice constant will introduce an error in the extracted phase: $\Delta\phi' \neq \Delta\phi$. For an accurate determination of the phase jump, the slope (producing the phase accumulation $\delta\phi$) thus needs to be minimised. **b** Constant current image (dashed box of Fig. 1b). **c** Fast Fourier transform of **b**, the insets show the two Bragg peaks used for the phase analysis. Note that Bragg peak 2 is clearly split. **d**, **e** Phase images from the two Bragg peaks. The insets show the total standard deviation of the phase ($\varepsilon_{\phi\alpha}$, see

text) for reference lattices spanning the dashed areas of the insets of **c**. The phase images are the optimised ones obtained from the reference lattice, which minimises $\varepsilon_{\phi\alpha}$ (white cross). **f** Magnitude of the displacement shift vector ($\bar{\phi}_1^R - \bar{\phi}_1^L, \bar{\phi}_2^R - \bar{\phi}_2^L$) between the two domains (see text) and its angle with respect to the 1D defect for a large number of 1D defects (both ridge- and dip-like). The phase shift of the ridge-like 1D defect (blue diamonds) is, on average, $\pi/2$ and predominantly oriented perpendicularly to the defect. The typical error bars, shown for the point extracted from **b**, predominantly reflect random phase fluctuations on both sides of the defect (see also Supplementary Note 2).

on each of the domains $L$, $R$ in absence of drift during the experiment (see Fig. 2d, e and Supplementary Information section 1 and 2). Hence the displacement phase shift ($\Delta\phi_1$, $\Delta\phi_2$) across the defect is accurately given by the difference of the phase averaged on each domain, $\Delta\phi_\alpha \equiv \bar{\phi}_\alpha^R - \bar{\phi}_\alpha^L$. Additionally, we can extract the angle of the 1D defect with respect to the atomic lattice, and thus the angle between the displacement shift and the 1D defect. Figure 2f shows the magnitude of the displacement phase, $\Delta\phi \equiv \sqrt{(\Delta\phi_1)^2 + (\Delta\phi_2)^2}$, and its angle to the defect extracted for nearly 100 topographies taken on different locations of different (ridge-like) 1D defects in Fig. 1a. Two observations stand out: the phase shift magnitude is, on average $\pi/2$ (corresponding to a quarter of a unit-cell), and the displacement tends to be perpendicular to the 1D defect. We note that the data of ref. 25 falls perfectly on top of our data when the above phase-slope optimisation method is applied (we recover their sloped results using integer pixel reference lattices). For comparison, we performed the same analysis on several locations on a valley-like 1D defect (green arrow of Fig. 1a), which has the same phase shift magnitude, but has its orientation along the defect instead of perpendicular to it, see also Supplementary Information section 3. Finally, we note that the same displacement phase shifts are obtained if the real-space fitting method introduced by ref. 33 is applied, see Supplementary Information section 1.

### Surface debris, endpoints, and gapping
The seeming absence of a $\pi$ lattice phase shift poses a challenge: could a non-quantised and hence non-topological value of the lattice phase shift still induce a quantised $\pi$ phase shift in the superconductor, and, if not, is there an alternative explanation for the near-constant in-gap

density of states along the 1D defect? While searching for an answer to these questions, we encountered several examples of unexpected behaviour. The first one is the presence of debris that forms 1D-like structures on top of the surface, see Fig. 3a. Although the surface debris is not accompanied by a phase shift of the lattice, it shows nearly identical behaviour in differential conductance (Fig. 3b, c and Supplementary Information section 4). Interestingly, the surface debris is often located close to an endpoint of a 1D defect, suggesting a possible link. Lastly, on continuous 1D defects, with a phase shift of order $\pi/2$ along the length of the defect, we find several short segments with a recovered gap in the density of states, see Fig. 3d, e and Supplementary Information section 5. Similarly, the dip-like 1D feature in Fig. 1a, which has a phase shift of similar magnitude as the ridge-like 1D defect, has little to no in-gap filling, see Supplementary Information section 3.

### Manipulating 1D defects
Before discussing the implications of these findings, we first focus on the surface debris and its possible relation to the 1D defects. One of the challenges with any type of adatom or debris is that it is relatively easily moved and/or picked up by the STM tip. During our studies, in particular, of the area in Fig. 3a, it proved very difficult to complete a spectroscopic map without modifying the debris and/or tip, suggesting that the debris is rather sensitive to the electric field of the tip. Curiously, we also observed small instabilities in the tunnelling signal on parts of the ridge-like 1D defects, although the tip and surface themselves did not seem to be modified, see Supplementary Information section 6 for examples.

To explore the instabilities in more detail, we scanned at increasingly smaller junction resistances (i.e., smaller tip-sample

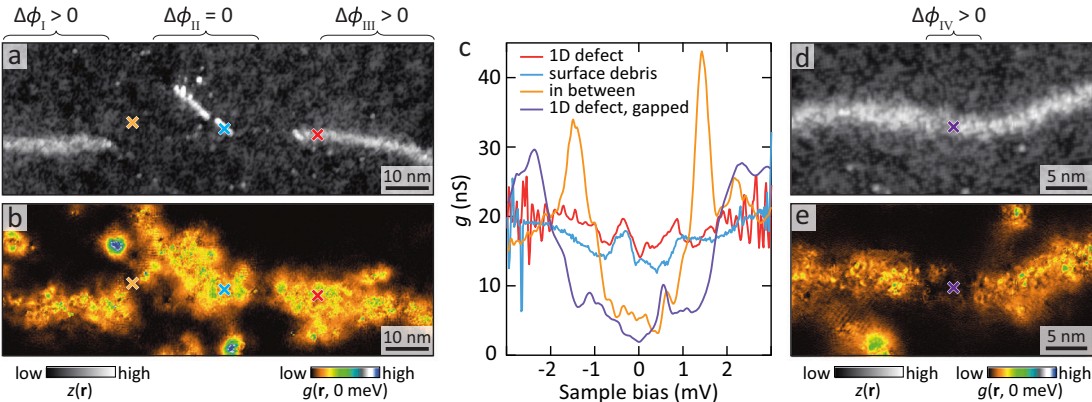

**Fig. 3 | End points, debris, and interruptions. a** Constant current image of a split 1D defect. In between the two segments is a strip of (1D) surface debris. $V = 5$ mV, $I = 100$ pA. **b** $g(E = 0)$ taken simultaneously with **a**. Even though there is within error no phase shift of the lattice at the surface debris area ($\Delta\phi_{II} = 0.14\pi \pm 0.19\pi$), the in-gap density of states is indistinguishable from that of the 1D defect (that has non-zero phase shifts $\Delta\phi_I = 0.52\pi \pm 0.12\pi$ and $\Delta\phi_{III} = 0.54\pi \pm 0.16\pi$). **c** Spectra marked with crosses in **a** on the 1D defect, the surface debris, and in between the two. **d** Constant current image and **e** $g(E = 0)$ on another 1D defect. Setup: $V = 5$ mV, $I = 80$ pA. Even though the phase shift of the lattice is constant along the 1D defect ($\Delta\phi_{VI} = 0.38\pi \pm 0.25\pi$), the differential conductance shows a gap. The cross marks the location of the spectrum in **c**.

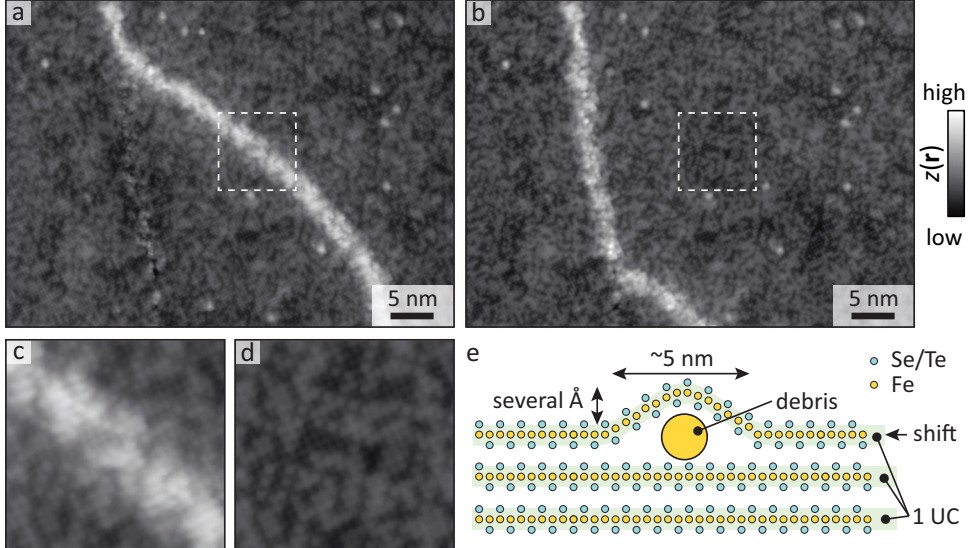

**Fig. 4 | Defect manipulation.** Constant current image of a 1D defect before (**a**) and after (**b**) manipulating it with the electric field of the tip. $V = 5$ mV, $I = 80$ pA. **c**, **d** Area in the dashed boxes of **a**, **b**, i.e., with and without defect, respectively. All atoms are clearly accounted for; they are simply pushed upwards in **c**. **e** Schematic of the 1D defect: debris (likely Fe clusters) underneath the surface warps the top unit cell (UC) layer, leading to a small shift of the lattice. The size of the debris is exaggerated for clarity.

distance meaning higher electric field) on a 1D defect where instabilities occurred. Occasionally, the 1D defect seemed to slightly change orientation before snapping back, until at sufficiently low junction resistance, it was permanently displaced to a new location, see Fig. 4a, b. Subsequent treatments enabled us to relocate additional sections of the 1D defect. The differential conductance before and after manipulation clearly shows the strong link between the defect and the enhanced in-gap signal (see Supplementary Information section 6 for more details). More importantly, we can now resolve the atomic lattice where the 1D defect used to be, as well as directly confirm that the lattice shifts only across the 1D defect. Interestingly, as Fig. 4c, d show, the lattice on top of the 1D defect and the same area after moving it away look very similar. In fact, all surface atoms seem to be accounted for, they are simply somewhat closer together on the 1D defect. Combined with the elevated height of the 1D defect, this strongly suggests that the surface is warped over a sub-surface defect as schematically depicted in Fig. 4e. This naturally explains the phase

shift of the lattice and its preferential direction: it shifts perpendicularly towards the defect to accommodate the warping. Using the observed height and width of the defect, the lateral shift is estimated to be on the order of an Ångstrom, which is the same order of magnitude as the shift and corresponding phase difference observed in experiment.

All findings combined strongly suggest that the 1D defects are debris buried just below the surface (Fig. 4e). Although the nature of the debris is hard to determine, a possible candidate is Fe aggregations. Fe atoms are known to create long chains on Pb[6] under conditions that may also have been met during the growth of our single crystals.

## Discussion

Can surface and/or sub-surface debris host 1D Majorana modes? From our analysis, one of the essential requirements for 1D dispersing Majorana modes, namely a $\pi$-phase shift of the superconductor, is

unlikely met: the lattice itself does not have such a quantised topological shift, so there is no reason to believe the superconductor would. Moreover, the gaps in the density of states on small portions of a defect with a near-constant lattice phase shift (Fig. 3d, e) strongly argue against the presence of topologically protected modes. If the debris has the correct spin texture, however, it may host Majorana bound states at the ends of the defects[5,7,10], whereas the defect itself would host conventional Yu-Shiba-Rusinov bands[34]. We find, however, no evidence of unusual behaviour at the endpoints of our 1D defects, see Fig. 3a and Supplementary Information section 7, suggesting that Majorana bound states are unlikely to be present. We, therefore, conclude that the gap filling at these 1D defects is topologically trivial in origin: strong scattering at clusters of impurities suppresses superconductivity and fills up the sub-gap region with states. The observation that local damage from heavy ion irradiation shows remarkably similar gap filling to that observed on 1D defects[29] further indicates that this is not an unreasonable scenario. More generally, our results show that a roughly constant sub-gap density of states is not a unique signature of linearly dispersing modes of a topological superconductor but, in fact, quite common for conventional mechanisms of gap-filling. Future studies at genuine crystalline domain walls may shed more light on whether or not topologically protected sub-gap states can truly be hosted by Fe(Se,Te).

## Methods

Fe(Se,Te) single crystals were grown using the self-flux method. As-grown samples with a superconducting transition temperature of 14.5 K were used throughout this work. The crystals were mechanically cleaved in cryogenic vacuum at $T \sim 20$ K and directly inserted into the STM head at 4.2 K. An etched tungsten tip was used for all measurements. Differential conductance measurements were performed by numerical derivation as well as with a lock-in amplifier operating at 429.7 Hz. All measurements were recorded at the base temperature of $T = 0.3$ K.

## Data availability

All raw data generated during the study are available from the corresponding authors upon request. The data shown in Fig. 2 and Supplemental Figs. S1–S3 are provided in the Source Data file. Source data are provided with this paper.

## Code availability

The computer code for extracting the phases uses standard Fourier analysis tools of imaging software, and is available upon request.

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

## Acknowledgements
F.M. would like to acknowledge funding from the ANR (ANR-21-CE30-0017-01). The work at BNL was supported by the US Department of Energy, Office of Basic Energy Sciences, contract no. DOE-sc0012704.

## Author contributions
F.M. performed the experiments. F.M. and A.M. analysed the data, interpreted the results and wrote the manuscript. G.D.G. provided the samples.

## Competing interests
The authors declare no competing interests.
