## [Peer Review File · Nature Communications]

REVIEWER COMMENTS

Reviewer #1 (Remarks to the Author):

The manuscript submitted to Nature Communications entitled “*Topologically trivial gap-filling in superconducting Fe(Se,Te) by sub-surface impurity chains*” by Mesaros *et al.* reports a detailed analysis of the phase shift induced by 1D defects observed on the surface of Fe(Te, Se). With the statistics they made, they found for 1D ridge like defects, the average phase shifts of the lattices between the two domains have a value near $\pi/2$. By checking the differential conductance mapping images at those defects, they found enhanced density of states with topological trivial origin at zero energy and near zero energy along the 1D defect structures. This result is in great contrast to one recent study (Science 367,104-108(2020)) where those in-gaps states were interpreted as evidence for dispersing Majorana mode. Moreover, they demonstrate the ability to manipulate one type of the defects (called debris), although only one manipulation is permanent.

The experimental results presented in the manuscript will be very interesting for readers who work in the fields like iron-based topological superconductor and helical (chiral) Majorana mode as well as stimulate further studies on the identification of dispersing Majorana mode on the surface of Fe(Te, Se). However, on the other hand, potentially it will also raise some debates on the nature of dispersing Majorana mode observed before. Therefore, it is very important to make all the discussions more solid. In my view, there are still several aspects can be improved before I can recommend its publication on Nature Communications. The main concerns and comments are listed below:

Main concerns:

1. From all the STM images the authors presented in both main text and supplemental material, quite a lot of adatoms can be identified near both kind of 1D defects. As these adatoms also show in-gap states, I assume those are the previously observed excess Fe atoms on the surface. In this sense, I have a feeling that the sample under study is kind of “dirty”. One natural question is that how the distribution of the Fe impurities is related to the observed 1D defect? For example, do the regions showing the 1D defect have more surface Fe impurities coverage compared to the regions without any 1D defect? If so, what is the reason for this? In what degree the impurities near or directly on the 1D defects affect the density of state in the mapping images? *Does the 1D defect in the impurities free regions showing similar in-gap density of state filling behavior compared to the results presented here in the manuscript?* I believe an in-depth discussion of the Fe impurities near the 1D defects will help for the understanding of the 1D defects formation mechanism as well as the origin of the in-gap state.

2. One of the main purposes of the experiments described in the manuscript is to check that whether the exact phase shift value and direction have a direct link with the appearance of in-gap states on the 1D defect. From Figure 2(f) we learned the authors measured lots of 1D defects with different phase shift values, but in the manuscript and supplemental material, only 0 and $\pi/2$ phase shift 1D defects are discussed. How the in-gap states look like for 1D defects with other phase shift values? Even there is no direct link between phase shift value and the appearance of in-gap states, it is worthy to put those results in the manuscript or supplemental material. Thus, the readers will get a full picture about the phase shift-in gap states relationship.

Other Comments

3. The title: Topologically trivial gap-filling in superconducting Fe(Se,Te) by sub-surface impurity chains

Although the structure is kind of one dimensional, however, the impurity below does not have to be chains. Maybe the following is another choice: “Topologically trivial gap-filling in superconducting Fe(Se,Te) by sub-surface impurities under one dimensional defects”

4. Page 1: “This insensitivity includes most types of random disorder, and as such makes topologically protected states the ideal platform for information transport without loss, either for e.g. spin-polarized states in the gap of an insulator, or Majorana modes inside a superconducting gap. “

I understand the authors want to say the information transformation is insensitive to most random impurities, which is the advantage of the topological protected defect state, however the current sentence is not so clear to the readers, it is better to rewrite this sentence.

5. Page 2, last sentence “We note that despite the presence of a small concentration of excess Fe atoms, the tunneling spectra away from the defect are fully gapped with gap sizes in agreement to those reported”. However, in Figure 1 d, I found the off-defect region shows a SC gap smaller than 1 meV which is not in agreement with previous reports. Although it is known as a multiband superconductor, the superconducting gap reported before usually higher than 1 meV (1.7 meV in ref. Science 328, 474 (2010), 1.8 meV in ref. Science 362, 333(2018), 1.4 meV in ref. Science 367, 104 (2020), it is also smaller compared to the Figure 1B in the ref. Sci. Adv.1, e1500033 (2015) by the same author). What is the reason for the SC gap reduction? Is this related to the high Fe adatoms coverage?

6. Is it a necessary to repeat the plot of Figure 1b in Figure 2b?

7. For the determination of the reference lattice, I understand the author try to minimize the slope of phase accumulation induced by a random chosen reference lattice. I wonder is this the best way to extract the real phase jump from left domain to right domain. To me, this procedure is a bit misleading. For example, if the authors want to extract the K value (let's note K^L) best fits the left lattice then one just needs to find the K point with a minimum $\varepsilon_{\phi_{1,2}}^L$. Then for the left lattice, the slope term is minimized. With the same procedure, one can also get the best fits lattice with K value K^R with a minimum $\varepsilon_{\phi_{1,2}}^R$. Thus, with these two exact K^L and K^R why not choose one of them as the reference lattice? To me this choice is straightforward, because if you use the right domain lattice as the reference lattice then the phase shift near the GB is totally introduced by the appearance of the left domain lattice. This is also used in the ref. 31 (section 6.1, first paragraph). Of course, there will always be a linear phase slope due to the difference between K^L and K^R which is $\Delta K \cdot r$ (like the images shown in Figure 2(e) and (f) in the ref. 31). However, we just need to concern the phase jump at the grain boundary (instead a phase average among the whole left or right domain region) which is the value we are trying to calculate and the linear term is trivial to understand. I am just afraid, the global smallest ε_{ϕ_1} on one hand can minimum the global phase slope, one the other hand mask some physical information (like additional lattice shift by other effect other than the 1D defect).

8. Page 4, “to determine the optimal reference lattice, we calculate the lattice phase images $\phi_{\alpha}^{K_{\alpha}}(r)$ for a finely spaced set of values of K_{α} covering the experimental 4x4 pixel area centred on the brightest pixel in the Fourier transform (Fig. 2c).”

It is not obvious to the readers how these phase images are constructed from the FFT images, it is worthy to explain more the detail in main text or in the method section. In this way the reading will be smoother.

9. Page 4, “We note that the data of Ref. [5] falls perfectly on top of our data when the above phase-slope optimization method is applied.”

Can the author comment on the possible reason why Ref. [5] gave a π phase shift and the author instead calculated a $\pi/2$ phase shift? Is this mainly due to the minimize of the global phase slope? Does this imply the method used in Ref. [5] is not suitable in dealing with the phase calculation? Maybe it is better to extend a bit more here to summarize and compare the advantage and disadvantages regarding the method used in both the manuscript and Ref. [5].

10. In Figure 3c, the red (1D defect) and blue (surface debris) spectra seem very noisy above ± 2 meV, what is the possible reason for this?

Reviewer #2 (Remarks to the Author):

The authors performed spectroscopic-imaging STM experiments on Fe(Se,Te), focusing on the line-like structural defects that were previously argued to host 1D dispersing Majorana fermions. One of the key ingredients necessary to achieve 1D Majorana fermions on the Fe(Se,Te) surface is that the line-like structural defects should be a particular domain wall characterized by the half-unit-cell lattice mismatch between the neighboring domains. The authors have developed a new scheme to analyze the lattice mismatch quantitatively and have investigated many defects to obtain statistically meaningful results. Unlike the previous result, the authors found that the half-unit-cell lattice mismatch is absent. The authors also demonstrated that the line-like structural defect is mobile while the surface atomic structure remains essentially intact. These results indicate that the line-like structural defects are not a topological object but are associated with the subsurface debris that breaks superconductivity. The experiments were thoroughly done, and the results were reasonably analyzed. It is a pity that the line-like structural defects are trivial in the end, but there is no doubt that this careful work is an excellent example for experimentalists. I am happy to recommend the publication of this manuscript in *Nature Communications*.

AUTHOR'S RESPONSE TO REVIEWERS:

Reviewer #1

The manuscript submitted to Nature Communications entitled “Topologically trivial gap-filling in superconducting Fe(Se,Te) by sub-surface impurity chains” by Mesaros et al. reports a detailed analysis of the phase shift induced by 1D defects observed on the surface of Fe(Te, Se). With the statistics they made, they found for 1D ridge like defects, the average phase shifts of the lattices between the two domains have a value near $\pi/2$. By checking the differential conductance mapping images at those defects, they found enhanced density of states with topological trivial origin at zero energy and near zero energy along the 1D defect structures. This result is in great contrast to one recent study (Science 367,104-108(2020)) where those in-gaps states were interpreted as evidence for dispersing Majorana mode. Moreover, they demonstrate the ability to manipulate one type of the defects (called debris), although only one manipulation is permanent.

The experimental results presented in the manuscript will be very interesting for readers who work in the fields like iron-based topological superconductor and helical (chiral) Majorana mode as well as stimulate further studies on the identification of dispersing Majorana mode on the surface of Fe(Te, Se). However, on the other hand, potentially it will also raise some debates on the nature of dispersing Majorana mode observed before. Therefore, it is very important to make all the discussions more solid. In my view, there are still several aspects can be improved before I can recommend its publication on Nature Communications. The main concerns and comments are listed below:

We thank the Reviewer for their positive evaluation of our manuscript and their insightful comments.

Main concerns:

1. From all the STM images the authors presented in both main text and supplemental material, quite a lot of adatoms can be identified near both kind of 1D defects. As these adatoms also show in-gap states, I assume those are the previously observed excess Fe atoms on the surface. In this sense, I have a feeling that the sample under study is kind of “dirty”. One natural question is that how the distribution of the Fe impurities is related to the observed 1D defect? For example, do the regions showing the 1D defect have more surface Fe impurities coverage compared to the regions without any 1D defect? If so, what is the reason for this? In what degree the impurities near or directly on the 1D defects affect the density of state in the mapping images? Does the 1D defect in the impurities free regions showing similar in-gap density of state filling behavior compared to the results presented here in the manuscript? I believe an in-depth discussion of the Fe impurities near the 1D defects will help for the understanding of the 1D defects formation mechanism as well as the origin of the in-gap state.

The Reviewer is correct that the adatoms are excess Fe atoms, which produce well defined in-gap states for isolated impurities. Away from the impurities, however, the spectrum for low concentrations still sharply drops to zero conductance inside the gap as observed in e.g. Sci. Adv.1, e1500033 (2015). For increasing concentrations of excess Fe atoms their in-gap states start overlapping and will lead to a filling of the gap. An example of such a high concentration can be seen in PRB 80, 180507(R) (2009). In our case the concentration is still low enough to observe a well defined gap as Fig. 1d shows. On the other hand, wherever we find a number of excess Fe impurities relatively close to each other, their sub-gap states start overlapping leading to a signature that indeed looks quite similar to the density of states at the 1D defects (see e.g. Figs. S5, S9 and S10). This does not mean, however, that they have anything to do with the density of states at the 1D defects: the parts of 1D defect without excess Fe impurities look indistinguishable from the parts with excess Fe atoms nearby. This is in line with our

conclusion that the 1D defects are resulting from sub-surface debris: an extra scattering centre in the form of an excess Fe atom will just add a bit to the even larger disorder introduced by the debris.

Concerning the concentration of excess Fe atoms: we observe no systematic variation in their density, meaning that whether or not there is a 1D defect does not seem to be a factor in their distribution on the surface. We have added this information to the main text.

2. One of the main purposes of the experiments described in the manuscript is to check that whether the exact phase shift value and direction have a direct link with the appearance of in-gap states on the 1D defect. From Figure 2(f) we learned the authors measured lots of 1D defects with different phase shift values, but in the manuscript and supplemental material, only 0 and $\pi/2$ phase shift 1D defects are discussed. How the in-gap states look like for 1D defects with other phase shift values? Even there is no direct link between phase shift value and the appearance of in-gap states, it is worthy to put those results in the manuscript or supplemental material. Thus, the readers will get a full picture about the phase shift-in gap states relationship.

We apologise for the confusion. Our analysis on a large number of 1D defects shows that the phase shift indeed varies, but is on average $\pi/2$ (and, most importantly, never π). Without a 1D defect, there is no abrupt phase shift (note, drift and strain fields may add phase shifts that vary more smoothly in space, discussed later). To distinguish between a defect with a non-zero phase shift, which is on average $\pi/2$, and a feature with no phase shift at all, we referred to these two situations as "0" and " $\pi/2$ ", which does sound confusing. To avoid the misunderstanding, and to clarify that we do not mean exactly $\pi/2$, we have now replaced " $\pi/2$ " with " $\sim\pi/2$ " throughout the text and figures. We have also indicated the typical error bar for the phase values in Fig. 2f which predominantly reflects random phase fluctuations on either side of the defect (more on this below) and shows more clearly to what extent variations in the phase are significant. Regardless of the exact value of their phase shift, the density of states of all ridge-like 1D defects looks similar: we cannot think of a more meaningful way to visualize this than the many supplemental figures of zero bias conductance we already present, all of which correspond to points in Fig. 2f. What is perhaps even more important, the DOS of the 1D defects is indistinguishable from the DOS at 1D surface debris (see Fig. 3 and S5). At the same time, the DOS may exhibit small energy gaps along a section of a 1D defect (see Fig. 3 and Fig. S6) even if the abrupt phase shift (of $\sim\pi/2$) occurs along that section as it does along the rest of the defect.

Other Comments

3. The title: Topologically trivial gap-filling in superconducting Fe(Se,Te) by sub-surface impurity chains. Although the structure is kind of one dimensional, however, the impurity below does not have to be chains. Maybe the following is another choice: "Topologically trivial gap-filling in superconducting Fe(Se,Te) by sub-surface impurities under one dimensional defects"

We have changed the title to "Topologically trivial gap-filling in superconducting Fe(Se,Te) by one dimensional defects."

4. Page 1: "This insensitivity includes most types of random disorder, and as such makes topologically protected states the ideal platform for information transport without loss, either for e.g. spin-polarized states in the gap of an insulator, or Majorana modes inside a superconducting gap. "

I understand the authors want to say the information transformation is insensitive to most random impurities, which is the advantage of the topological protected defect state, however the current sentence is not so clear to the readers, it is better to rewrite this sentence.

We agree that this sentence was not very clear. In the revised text we have replaced it with: "This insensitivity includes most types of random disorder, and as such makes topologically protected states a promising platform for quantum information technology."

5. Page 2, last sentence "We note that despite the presence of a small concentration of excess Fe atoms, the tunneling spectra away from the defect are fully gapped with gap sizes in agreement to those reported". However, in Figure 1 d, I found the off-defect region shows a SC gap smaller than 1 meV which is not in agreement with previous reports. Although it is known as a multiband superconductor, the superconducting gap reported before usually higher than 1 meV (1.7 meV in ref. Science 328, 474 (2010), 1.8 meV in ref. Science 362, 333(2018), 1.4 meV in ref. Science 367, 104 (2020), it is also smaller compared to the Figure 1B in the ref. Sci. Adv.1, e1500033 (2015) by the same author). What is the reason for the SC gap reduction? Is this related to the high Fe adatoms coverage?

We thank the Reviewer for pointing out that the 'off-defect' spectrum in Fig. 1d is confusing. After double checking, this spectrum turns out to contain a sub-gap state, either from a sub-surface excess Fe atoms, or simply because it was still too close to the 1D defect. The second peak in the spectrum (located at ~1.5meV) is the coherence peak, which is in agreement with all previous studies. We have replaced this confusing spectrum with one taken a little further from the 1D defect in a region where there is no sub-gap state.

6. Is it a necessary to repeat the plot of Figure 1b in Figure 2b?

We were also considering this issue, and motivated by the Reviewer's comment we found a way to improve the presentation. First we emphasize that the purpose of Fig. 1 is to establish that the 1D defect we observe is identical to the one reported previously. The comparison between topography and zero bias conductance is therefore essential. Second, Fig. 2 on the other hand explains our approach to extract the phase difference. Since it is important to establish the logic of the approach, which is entirely based on the topography, the topograph should also be presented. To avoid repetition, we could have chosen a different region of the material to show in Fig. 1b,c, but again we believe that it is important for the reader to see the phase analysis on the same region where we show the conductance, so that all data are coherently aligned. Hence the improvement we found was in showing a larger field of view in Fig. 1b,c that entirely contains a smaller region (dashed box) in which the phase analysis is performed in Fig.2; this avoids duplication, but still allows the reader to see both the phase and the conductance on the same area of the sample.

7. For the determination of the reference lattice, I understand the author try to minimize the slope of phase accumulation induced by a random chosen reference lattice. I wonder is this the best way to extract the real phase jump from left domain to right domain. To me, this procedure is a bit misleading. For example, if the authors want to extract the K value (let's note K_L) best fits the left lattice then one just needs to find the K point with a minimum $\epsilon_{0,2L}$. Then for the left lattice, the slope term is minimized. With the same procedure, one can also get the best fits lattice with K value K_R with a minimum $\epsilon_{0,2R}$. Thus, with these two exact K_L and K_R why not choose one of them as the reference lattice? To me this choice is straightforward, because if you use the right domain lattice as the reference lattice then the phase shift near the GB is totally introduced by the appearance of the left domain lattice. This is also used in the ref. 31 (section 6.1, first paragraph). Of course, there will always be a linear phase slope due to the difference between K_L and K_R which is $\Delta K \cdot r$ (like the images shown in Figure 2(e) and (f) in the ref. 31). However, we just need to concern the phase jump at the grain boundary (instead a phase average among the whole left or right domain region) which is the value we are trying to calculate and the linear term is trivial to understand. I am just afraid, the global smallest

ϵ_0 on one hand can minimize the global phase slope, on the other hand mask some physical information (like additional lattice shift by other effect other than the 1D defect).

The Reviewer raises an interesting point. Let us start by stressing that in this work we are purely interested in the phase jump across the 1D defect and would like to avoid or compensate for possible additional smooth position-dependent changes in the phase. Crucially, since atomic contrast is absent at the 1D defect itself, we cannot extract the phase jump at the defect and are therefore forced to consider the phase some distance from the defect – and for this reason we have to minimize the slope. In fact, we must remove the area near the defect, as the loss of atomic contrast there can only provide spurious Fourier data. To increase the accuracy of the extracted phase jump, we hence must consider extended regions ("left" and "right") on both sides of the defect, as a priori neither is preferred. We agree then that one may ask how to extract the phase jump across defect by analyzing the lattices in one or both of the regions. In principle, the lattice in absence of defects has a single lattice constant. We have confirmed this on a number of topographs taken far from 1D defects. Based on this notion, using only one region (i.e. half of the field of view) to determine the optimal slope seems illogical, which is one reason why we added the standard deviations of both sides, i.e., we demanded that the determined optimal lattice is an equally good description in both regions, which are a priori indistinguishable. Using the optimal lattice, in both regions the phase should become as constant as possible (more precisely, fluctuate minimally and randomly around a fixed average), and hence the phase jump across the defect is just the difference of these constant values on the right and left region.

The Reviewer rightly wonders if there could be additional global phase variations across both regions that could affect the reliability of our phase-jump-extraction method. First, imagine that there is an additional linear increase in the phase across both regions, i.e., due to a uniform uni-directional strain. This contribution, being a linear function of position, would simply change the optimal lattice parameter, but would not affect the determined constant phase value on each region, and hence would not affect the extracted phase jump across the defect. Hence, only non-linear changes to the lattice constant are important. There are two physical possibilities for such non-linearity: drift and (non-uniform) strain.

Drift, due to the slow (temperature dependent) relaxation of the piezos after a change in voltage, will stretch or compress an image in the direction of the drift. In our measurements, we have tried to minimize non-linear drift by staying in the same location for extended periods of time, waiting sufficiently long after a temperature change and scanning slowly. Nevertheless, non-linear drift could still appear, for example in the first image after a temperature change. This, however, always shows up at the start of the image and in the slow scan direction (the y-axis in all our data) and as such can easily be recognized. For vertically running 1D defects, using the left (L) or right (R) region only will thus not make a difference in this case. Nevertheless, whenever we did observe non-linear drift, we have not taken the affected part of the image into consideration.

A more interesting case is when the lattice constant itself is changing non-linearly as function of position due to strain (as discussed in Ref. 31 and for example exploited with STM in PNAS 115, 6986-6990 (2018)). Again we stress that a linear strain-field will be corrected for by our optimization method. A non-linear strain field, however, will give a different slope of the phase for the left and right region of the image. Let us consider the situation where the lattice parameter is constant on the left, progressively stretched on the right with a positive jump from left to right. Optimizing our reference lattice to the left side will give a constant phase on the left, a positive jump followed by a phase with a positive slope on the right. As Fig. 2a shows, this will lead to an overestimation of the jump in the phase. Conversely, if we optimize on the right, we will underestimate the jump. The most reliable phase jump is thus extracted when taking both sides into account. We have added a section to the Supplementary

Information (section 2) to discuss this point, including also a figure showing the effect on data taken near strain fields and far from it.

To best clarify the interplay of our phase-optimization approach and the physical quantities such as phase-jump and non-linear strain, in Fig. S2 we include error bars on the extracted values of the phase jump. These error bars represent the standard deviation due to the random spatial variation of the phase field across both regions. Although the error bars include some of the uncertainty due to the spatial fluctuations of strain, they do not at all contain the systematic error made by the method one chooses to remove the underlying smooth non-linear strain profile. As the example in the bottom row of Fig. S2 demonstrates, when we extract the phase jump using three methods, namely, considering only the R region, only the L region, or both, we get three values of phase jump magnitude which agree within the statistical error bars. This is strong indication that non-linear strain is negligible in this example, i.e. we do not have a systematic error due to strain. The example in top row of Fig. S2 produces three values of phase jump that do not overlap within the statistical error bars. Hence, we know that the three methods (R only, L only, or both) give three distinguishable reference lattices which compensate differently for the non-linear strain. As argued in previous paragraph, the method using both L and R is the one that minimizes the bias in the phase jump (and it gives a value in-between the ones given by only L or only R).

We note that in the process of determining the error bars, we realized our original addition of standard deviations to extract the minimal slope in the phase actually introduces a small systematic error. This is because the size of the L and R areas is not necessarily identical. If one simply uses $\bar{\epsilon} = \epsilon_L + \epsilon_R$, the smaller of the two sides will get slightly more weight, because $\bar{\epsilon} = \frac{\sum_i (x_i - \bar{x}_L)^2}{N_L} + \frac{\sum_j (x_j - \bar{x}_R)^2}{N_R} = \frac{N_R \sum_i (x_i - \bar{x}_L)^2 + N_L \sum_j (x_j - \bar{x}_R)^2}{N_L N_R}$. If region L is very small, N_L is very small, and thus the standard deviation of region R hardly contributes. To avoid this problem, we now use a weighted summation, namely: $\bar{\epsilon} = \frac{N_L \epsilon_L + N_R \epsilon_R}{N_L + N_R}$. We stress that since nearly all fields of view we use in our study have roughly equal L and R areas, the change in the extracted phase jump and orientation is minimal.

The new section 2 of the Supplementary Information presents the arguments in the above paragraphs.

8. Page 4, “to determine the optimal reference lattice, we calculate the lattice phase images $\phi_{\alpha} K_{\alpha}(r)$ for a finely spaced set of values of K_{α} covering the experimental 4x4 pixel area centred on the brightest pixel in the Fourier transform (Fig. 2c).”

It is not obvious to the readers how these phase images are constructed from the FFT images, it is worthy to explain more the detail in main text or in the method section. In this way the reading will be smoother.

We thank the Reviewer for their suggestion, and we have added a short description of the basic principle of the technique in the main text. We stress that we use the exact same technique as in Refs. 32 and 5 and refer to these papers for more details. We additionally emphasize in the main text and Supplementary Information that our crucial novelty is to extend the analysis to consider non-integer pixel values of the reference lattice Bragg vector. The analysis generally allows for this, but thus far in the literature only integer pixel values were used for convenience.

9. Page 4, “We note that the data of Ref. [5] falls perfectly on top of our data when the above phase-slope optimization method is applied.”

Can the author comment on the possible reason why Ref. [5] gave a π phase shift and the author instead calculated a $\pi/2$ phase shift? Is this mainly due to the minimize of the global phase slope? Does this imply the method used in Ref. [5] is not suitable in dealing with the phase calculation? Maybe it is better to extend a bit more here to summarize and compare the advantage and disadvantages regarding the method used in both the manuscript and Ref. [5].

This is a crucial point. As we show in Fig. 2 and discuss in detail in Supplementary Information section 1, a reliable phase difference can only be extracted if the background slope is minimised. This means that the ideal lattice constant needs to be used to extract the phase, i.e. one must not constrain the Bragg vector to integer pixels in the Fourier transform. (For completeness, let us repeat that the real-space image, of size N pixels in a certain direction, does not generally cover a precisely integer number of unit-cells, each being of size " a ". Hence, in Fourier space, the Bragg peak with value given by $1/a$ will not be an integer multiple of the pixel in Fourier space given by $1/N$.) The phase analysis technique itself is identical to that used in Ref. [5], except that we use the ideal lattice constant defined by a Bragg vector which may have any non-integer value. In other words, if we analyse the data of Ref. [5] using an integer pixel reference lattice, we recover the results presented in Ref. [5]. We now added this information to the main text. Due to the background slope, however, the phase shift obtained with integer-valued Bragg peaks is obviously not correct. With the ideal lattice, the obtained phase shift is reliable and falls on top of our data. The situation is analogous to the example in Fig. S1 of Supplementary, where a striking difference occurs between top and middle rows, highlighting the importance of considering non-integer pixel values in the (otherwise identical) analysis.

In Supplementary section 2 we added a final paragraph that emphasizes how using a strictly integer-valued reference-lattice Bragg peak is a dangerous systematic error in the determination of phase-jump, since it affects all phase jumps (unrelated to non-linear strain), and may drastically bias the phase jump simply because a shift of K_α by half a pixel corresponds to a spurious accumulation of phase of order π across the field of view. Revisiting the basic mechanism in Fig.2a, one sees that a spurious accumulation of π across the field-of-view may easily push the extracted phase jump at the defect from $\sim\pi/2$ to $\sim\pi$.

10. In Figure 3c, the red (1D defect) and blue (surface debris) spectra seem very noisy above ± 2 meV, what is the possible reason for this?

We appreciate the attentiveness of the Reviewer. The reason is most likely related to the occasional shifting of the 1D defect (see Fig. S7). In all regions where there are no ridge-like 1D defects (or surface debris) the spectra show little noise. On top of many ridge-like 1D defects as well as on surface debris, however, the noise is enhanced for certain values of the voltage. In fact, for ridge-like 1D defects we find locations where bi-stability can be observed: at a fixed tip-sample distance the current fluctuates between two values. In some cases we can even link the two current values to slight differences in topography. This suggests that although the global arrangement of the sub-surface debris is mostly static (i.e. it is hard to move an entire section of defect), it can still locally move in response to the tip or current. We have added discussion of this point to Supplementary Information section 7.

Reviewer #2 (Remarks to the Author):

The authors performed spectroscopic-imaging STM experiments on Fe(Se,Te), focusing on the line-like structural defects that were previously argued to host 1D dispersing Majorana fermions. One of the key ingredients necessary to achieve 1D Majorana fermions on the Fe(Se,Te) surface is that the line-like structural defects should be a particular domain wall characterized by the half-unit-cell lattice mismatch

between the neighboring domains. The authors have developed a new scheme to analyze the lattice mismatch quantitatively and have investigated many defects to obtain statistically meaningful results. Unlike the previous result, the authors found that the half-unit-cell lattice mismatch is absent. The authors also demonstrated that the line-like structural defect is mobile while the surface atomic structure remains essentially intact. These results indicate that the line-like structural defects are not a topological object but are associated with the subsurface debris that breaks superconductivity. The experiments were thoroughly done, and the results were reasonably analyzed. It is a pity that the line-like structural defects are trivial in the end, but there is no doubt that this careful work is an excellent example for experimentalists. I am happy to recommend the publication of this manuscript in Nature Communications.

We thank the Reviewer for their positive evaluation of our manuscript.

REVIEWER COMMENTS:

Reviewer #1 (Remarks to the Author) :

I thank the authors very much for their efforts addressing my comments and the improvement of both the writings and figures based on my suggestions. Most of my questions have already been answered satisfactorily and I am happy to recommend this manuscript's publication on *Nature Communications*. I believe the findings by the authors will attract research attentions and stimulate experiments for re-checking the nature of previously observed "Dispersing Majorana mode" in different material systems. There are still two minor remarks from my side for the authors' considerations to make this manuscript even stronger.

1. I suggest label all the dI/dV mapping images both in manuscript and supporting materials with the exact phase shift values which already included in Fig. 2f instead of " $\sim\pi/2$ " (which will be more precise).

2. I think finally I understand why the authors use a reference lattice optimized both from the left and right side. The key is the losing atomic resolution at the grain boundary area. So a direct check for the phase jump at the grain boundary no longer accessible. Thus, to minimize the phase accumulation one has to optimize both sides. Maybe the author can point this out in the main text.

AUTHOR'S RESPONSE TO REVIEWERS:

Reviewer #1

I thank the authors very much for their efforts addressing my comments and the improvement of both the writings and figures based on my suggestions. Most of my questions have already been answered satisfactorily and I am happy to recommend this manuscript's publication on Nature Communications. I believe the findings by the authors will attract research attentions and stimulate experiments for re-checking the nature of previously observed "Dispersing Majorana mode" in different material systems. There are still two minor remarks from my side for the authors' considerations to make this manuscript even stronger.

1. I suggest label all the dI/dV mapping images both in manuscript and supporting materials with the exact phase shift values which already included in Fig. 2f instead of " $\sim\pi/2$ " (which will be more precise).

We have inserted the actual values (with errors) of the phase jump. In particular, this concerns Fig. 3, and Fig. S3. We left a mention of $\sim\pi/2$ in the main text where it does not refer to a specific observation, but instead to a general value that well characterizes the many observed instances of non-zero phase jump.

2. I think finally I understand why the authors use a reference lattice optimized both from the left and right side. The key is the losing atomic resolution at the grain boundary area. So a direct check for the phase jump at the grain boundary no longer accessible. Thus, to minimize the phase accumulation one has to optimize both sides. Maybe the author can point this out in the main text.

Our revised manuscript includes this argument in the 4th paragraph of the Results section of the main text (page 6), we quote:

"The key method change with respect to Ref. [32] is that although a considered $K\alpha$ is fixed for the entire image, the standard deviations of the two domains L, R are separately determined and then added up - without considering the region where the phase jump occurs. This is because we are aiming to determine the periodicity of the ideal lattice, which we assume to be identical on either side of the defect (see also Supplementary Note 2). Including the phase jump $\Delta\phi$ (which we cannot locally extract due to absence of atomic contrast) would erroneously increase the total slope of the phase across the image by $\Delta\phi/l$, where l is the image length...".

The Supplementary Note 2 referenced in the above text goes into further details and tests of the arguments, and contains the entirety of our reply to the Reviewer concerning this subject.

We do not see how we could better introduce and emphasize in the main text our argument about using domains on both sides of the defect without overly repeating ourselves.